# Caffeic Acid-Zinc Basic Salt/Chitosan Nanohybrid Possesses Controlled Release Properties and Exhibits In Vivo Anti-Inflammatory Activities

**DOI:** 10.3390/molecules28134973

**Published:** 2023-06-24

**Authors:** Carla Carolina Ferreira Meneses, Paulo Robson Monteiro de Sousa, Kely Campos Navegantes Lima, Lisa Maria Mendes de Almeida Souza, Waldeci Paraguassu Feio, Cláudio Márcio Rocha Remédios, Jenny Jouin, Philippe Thomas, Olivier Masson, Cláudio Nahum Alves, Jerônimo Lameira, Marta Chagas Monteiro

**Affiliations:** 1Laboratório de Planejamento e Desenvolvimento de Fármacos, Instituto de Ciências Exatas e Naturais, Universidade Federal do Pará, Belem 66075-110, Pará, Brazil; carlacarolina.meneses@gmail.com (C.C.F.M.); nahum@ufpa.br (C.N.A.); lameira@ufpa.br (J.L.); 2Microbiology Laboratory, Faculty of Pharmacy, Federal University of Pará, Belem 66075-110, Pará, Brazil; kcnavelima@gmail.com (K.C.N.L.); lisasouza.farmacia@gmail.com (L.M.M.d.A.S.); martachagas@ufpa.br (M.C.M.); 3Programa de Pós-Graduação em Física, Federal University of Pará, Belem 66075-110, Pará, Brazil; paraguassu@ufpa.br (W.P.F.); remedios@ufpa.br (C.M.R.R.); 4Laboratoire IRCER, Université de Limoges-CNRS UMR 7315, Centre Européen de la Céramique, 87068 Limoges, France; jenny.jouin@unilim.fr (J.J.); philippe.thomas@unilim.fr (P.T.);

**Keywords:** caffeic acid, intercalation, ZBS-CA, ZBS-CA/Ch nanoparticles, anti-edematogenic

## Abstract

Caffeic acid (CA) exhibits a myriad of biological activities including cardioprotective action, antioxidant, antitumor, anti-inflammatory, and antimicrobial properties. On the other hand, CA presents low water solubility and poor bioavailability, which have limited its use for therapeutic applications. The objective of this study was to develop a nanohybrid of zinc basic salts (ZBS) and chitosan (Ch) containing CA (ZBS-CA/Ch) and evaluate its anti-edematogenic and antioxidant activity in dextran and carrageenan-induced paw edema model. The samples were obtained by coprecipitation method and characterized by X-ray diffraction, Fourier transform infrared (FT-IR), scanning electron microscope (SEM) and UV-visible spectroscopy. The release of caffeate anions from ZBS-CA and ZBS-CA/Ch is pH-dependent and is explained by a pseudo-second order kinetics model, with a linear correlation coefficient of R^2^ ≥ 0.99 at pH 4.8 and 7.4. The in vivo pharmacological assays showed excellent anti-edematogenic and antioxidant action of the ZBS-CA/Ch nanoparticle with slowly releases of caffeate anions in the tissue, leading to a prolongation of CA-induced anti-edematogenic and anti-inflammatory activities, as well as improving its inhibition or sequestration antioxidant action toward reactive species. Overall, this study highlighted the importance of ZBS-CA/Ch as an optimal drug carrier.

## 1. Introduction

Caffeic acid (CA) is a natural phenolic compound found in various plants, including coffee, fruits, and vegetables [1]. It has been studied for its potential anti-inflammatory properties [1,2]. However, this compound exhibits low water solubility, which can hinder its effective delivery and bioavailability [3]. Strategies to enhance its solubility include the nanoscale delivery systems increase surface area and improve dissolution [4,5]. These nanocarriers can also provide controlled release of CA, allowing for sustained and regulated delivery of the compound [6]. This controlled release can optimize its therapeutic or functional effects, ensuring that the desired concentration of drug is maintained over a desired period of time [7]. In addition, nanocarriers can be loaded with different types of anti-inflammatory drugs, such as nonsteroidal anti-inflammatory drugs (NSAIDs) [8,9], corticosteroids [10], or specific anti-inflammatory peptides [11].

The Zinc Basic Salt (ZBS) have the general formula [Zn_5_(OH)_8_](A)_2_·*n*H_2_O], where A is an intercalated anion such as a carbonate, nitrate, carboxylate or sulfate group [12,13]. Their structures are related to the natural mineral Brucite and feature octahedral, edge-sharing sheets of zinc hydroxide units, where two tetrahedral zinc hydroxide units are situated above and below a vacant octahedral site [13,14]. The interlayer anions are weakly bound to the tetrahedral Zn^2+^ ions, completing the tetrahedral coordination, and easily exchanged with most of inorganic anions as well as organic ones [13,14].

The use of nanocarriers ZBS is not extensively reported in the literature compared to other drug delivery systems or applications. However, ZBS can be combined with chitosan (Ch), a biocompatible and biodegradable polysaccharide derived from chitin [15], to create composite materials or delivery systems with unique properties [16,17,18]. Consequently, the incorporation of nanoparticles, such as metal nanoparticles, silica nanoparticles, or polymeric nanoparticles, into Ch matrices can enhance drug loading capacity, provide controlled release profiles, and improve targeted delivery of therapeutic agents. Based on this, the combination of ZBS nanohybrids with Ch (ZBS-Ch) holds potential for various drug delivery routes [19].

This study shows for the first time the intercalation of the CA into inorganic layers (ZBS) complexed by Ch, ZBS-CA/Ch nanoparticles and in vivo pharmacological assays from anti-edematogenic and antioxidant in dextran and carrageenan-induced paw edema model. The motivation of this study is based on well-known biological activities of caffeic acid (CA), such as cardioprotective action [20], antioxidant [21], antitumor [22], anticarcinogenic [23], anti-inflammatory [24], and antimicrobial [25] among others. However, therapeutic applications of CA are still limited due to its low water solubility and poor bioavailability [26].

## 2. Results and Discussion

### 2.1. Powder X-ray Diffraction Pattern (PXRD) Analysis

The co-precipitation method was used to intercalate CA in the interlayer spacing of ZBS material, which were confirmed by XRD patterns (Figure 1). The interlayer spacing expands from 9.6 Å for the pristine ZBS-NO_3_ (Figure 1A) to 10.14 Å for the exchange product, ZBS-CA intercalation compound (Figure 1C). Upon exchange of the interlayer nitrate anions with caffeate anions, the layered nature of the pristine ZBS is preserved and the observed layer spacing of 10.5 Å is also close to 10.44 Å reported previously [27], observed from XRD analysis. The considerable broadening of the diffraction peaks up on intercalation of the organic anion (9.6 Å for ZBS-NO_3_ Figure 1A and 10.5 Å in the ZBS-CA Figure 1C) is often associated with a decrease in the particle size of the crystallites or turbostraticity of the layers whereby there is irregular or mismatched stacking of sequential layers [28]. Furthermore, the interlayer spacing of the 10.5 Å value suggests a gallery of the inorganic matrix caffeate anions assembled in bilayers. Such arrangement maximizes π-π interactions between the interlayer anions, hydrogen bonding interaction between the layer OH groups and the phenolic OH groups of the anions as well as the interaction between the layers and the anions as proposed by reference [27].

Additionally, the scanning electron microscope confirm a decreasing in crystallite size of the pristine ZBS to ZBS-CA (Appendix A). The pristine ZBS consists of plate-like particles with diameter ranging from 100 to 500 nm whilst the particles of the exchange product appear as aggregates of different shapes and sizes.

### 2.2. Infrared Analysis

The FT-IR spectra of all the samples are shown in Figure 2, however only the main absorption bands, between 1800–400 cm^−1^ spectral region, are shown for the sake of clarity. For the ZBS-NO_3_ sample (Figure 2A), the most intense absorption band is found at 1384 cm^−1^, which is characteristic of the free interlayer nitrate group (belonging to point group symmetry D_3h_) and at low frequency, bands arising from the intra-layers vibrations at 637 cm^−1^ of Zn–O and 467 cm^−1^ of O–Zn–O are detected, respectively as reported by [29]. Furthermore, the most notable feature upon intercalation of CA molecules is the disappearance of the absorption peak at 1643 cm^−1^ that is associated with stretching vibrations of the undissociated carboxylic acid (COOH) (Figure 2B) and the appearance of absorption peaks at 1545 cm^−1^ and 1421 cm^−1^ which confirms the intercalation of the caffeate anion between interlayer space of the ZBS, ZBS-CA sample (Figure 2C). In ZBS-NO_3_ (Figure 2A), the carboxylate group may coordinate with the ZBS cation as a unidentate ligand or occur as a free species [29]. However, sodium salts of the caffeate anion exhibit unidentate coordination (Na^+^CA^−^: 146 cm^−1^) [27], and a value similar obtained in this work by ∆*ni* = 1543–1421 = 122 cm^−1^, suggest that the interaction of the caffeate anion with the matrix cation in the ZBS is fairly electrostatic, also confirmed by UV-vis solid analysis (see Figure 3B). Finally, the FT-IR spectra of pure Chitosan (Figure 2D), show three characteristic absorption bands at 1657, 1561 and 1317 cm^−1^ which assigned to C=O stretching (amide I), N–H bending (amide II) and C–N stretching (amide III) modes of the residual N-acetyl groups, respectively, which remain after of the complexation of the ZBS-CA, Figure 2E.

### 2.3. Interpretation of the UV-Visible Absorption Spectra

The strength of the interaction between CA molecules and the ZBS lattice was analyzed by UV–Vis spectroscopy. Figure 3A shows four absorption bands at 215, 240, 287 and 312 nm which correspond to the wavelengths maximus (*λ_max_*) of the pure caffeic acid and released CA from the ZBS-CA and ZBS-CA/Ch samples. These values correspond to HOMO (highest occupied molecular orbital) for LUMO (lowest unoccupied molecular orbital) electronic transitions, for example, $n\to \pi^{*}$ and $\pi \to \pi^{*}$, that are attributed to carbonyl group (at 312, 287 and 215 nm) and aromatic ring (at 240 nm) in the caffeic acid molecule. Therefore, the data are composed of three UV bands, i.e., UV-A (315–400 nm), UV-B (280–315 nm) and UV-C (200–280 nm), which demonstrate that there is no degradation of the molecule in the synthesis process. Furthermore, an amount of the 46% (*w*/*w*) of the cafeilate anion intercalated into ZBS (ZBS-CA) was obtained, which is in compliance with 47.4% (*w*/*w*) reported in the literature [27].

Furthermore, the 46% (*w*/*w*) of the cafeilate anion of the ZBS-CA sample keep up 28 wt% of the CA into ZBS-CA/Ch sample after the complexation in Chitosan. The loss of the 18 wt% can be explained by CA molecules adsorbed in interlayer space of the ZBS as well as by the time (4 h) of the reaction of the ZBS-CA with Ch solution. A minor time could lead to the permanence of the CA molecule into ZBS-CA/Ch complex. Besides, the UV-Vis solid state spectrum (Figure 3B) shows peaks at 384 nm and 390 nm related to the energy transitions of the π-π to π→π^∗^ interactions, which correspond CA alone and of the caffeilate anion into ZBS layers, respectively.

### 2.4. Drug Release Behavior of ZBS-CA and ZBS-CA/Ch

The drug release properties of caffeic acid (CA) anion from the ZBS-CA and ZBS-CA/Ch samples have been investigated at a constant volume and temperature of 37 ± 0.5 °C at different pH values, indicating that the release rate of CA anion from these hybrids materials is pH-dependent. Figure 4A shows the release profiles of composite in solution at pH 4.8 for ZBS-CA and ZBS-CA/Ch. The percentage release of CA anion from ZBS-CA and ZBS-CA/Ch is approximately 41% and 23% after 10 h when exposed to pH 7.4 and approximately 77% and 46% to pH 4.8, respectively (Table 1). The slower release rate for ZBS-CA/Ch is due to the protective shell around the CA anions promoted both by the inorganic material (ZBS) and polymer Ch. In addition, five kinetic models (Korsmeyer-Peppas, Higuchi, Zero-Order, First-Order and Pseudo-Second Order) were assigned to understand the release mechanism of CA anion from the interlayer spaces of ZBS-CA and ZBS-CA/Ch layered compounds.

According to the literature [30], the kinetic models for layered compound are well described by linear regression technique. The calculated linear correlation coefficient shows that the R^2^ > 0.99 in pseudo-second order is the best kinetics model fitted to explain the release of CA anion from the interlayer spaces of ZBS compared to the other kinetic models (Table 1). The zero-order and first-order models are not suitable to explain the whole release of hybrids nanomaterials, as reported in the literature [31]. However, a detailed examination of the data point distribution in Figure 4 in pH 4.8 suggests that the whole release process consists of two linear stages. In order to better simulate the CA release behavior, we applied the kinetic models as two separate stages, the stage I in 0–45 min, and the stage II in 45–600 min. The stages I and II for CA anions release from ZBS/CA and ZBS-CA/Ch are best fitted with the pseudo-second order model, with a linear correlation coefficient of R^2^ ≥ 0.99 (Table 1).

Additionally, the simulation results of the kinetic model suggest that (i) the release at both stages is diffusion-controlled; (ii) within the first 45 min (stage I), most caffeate anions on the surface of ZBS particles diffuse into the medium solution via anion exchange; and (iii) at stage II, surface diffusion, although not the controlling step, is continuous; the controlling step is the caffeate anion diffusion from the inside to the surface of ZBS particles, which takes a longer time than at stage I, as previously reported [32]. Furthermore, the release profile (Figure 4) shows similar features to those previously reported for layered inorganic materials, such as layered double hydroxides (LDH) [32,33,34] and zinc basic salts (ZBS) [27,35,36] intercalated with organic compounds. Therefore, the release profile suggests that the CA anion release from ZBS-CA and ZBS-CA/Ch may be attributed to heterogeneous diffusion from the flat surfaces via ion exchange and/or by intraparticle diffusion or surface diffusion, as previously reported [37].

### 2.5. In Vivo Pharmacological Assays

#### 2.5.1. Antiedematogenic and Antioxidant of ZBS-CA/Ch in Dextran-Induced Paw Edema Model

As shown in Figure 5A, subcutaneous injection of dextran induced paw edema that reached a maximum edematogenic peak at 60 min (2.448 ± 0.834 mm), returning to normal levels at 240 min. HDS (4 mg/Kg), CA (10 mg/Kg) and ZBS-CA/Ch (50 mg/Kg) inhibited the paw edema from 30 min after dextran injection, remaining with this inhibitory effect until at least 120 min (Figure 5C,D) compared to the DEX group. The antiedematogenic effect along time (AUC 30–120 min) was observed in Figure 5(B.1) (DEX: 189.8 ± 17.02%; HDS: 107.8 ± 4.53%; CA: 107.6 ± 9.13% and ZBS-CA/Ch: 104.6 ± 7.54%). On the other hand, after 150 min, only the animals treated with the micro particles (ZBS-CA/Ch group) showed reduced edema (AUC 150–240 min) compared to the DEX group (DEX: 131.0 ± 10.49% and ZBS-CA/Ch: 94.52 ± 7.17%) (Figure 5(B.2)).

Regarding the oxidative parameters, our data show that 240 min after the induction of paw edema with Dextran, the DEX group showed no change in TEAC levels (Figure 5E), but increased NO and MDA levels in the paw (NO: 69.33 ± 6.17 μM/L; MDA: 22.71 ± 2.43 nmol/L) compared to the baseline group (Figure 5F,G). Regarding the treatments, HDS, CA and ZBS-CA/Ch also did not change the levels of TEAC, but reduced the production of NO and MDA levels in the paw as compared to the DEX group (Figure 5F,G), However, the animals treated with the ZBS-CA/Ch nanoparticles showed a better inhibitory effect on the oxidative imbalance in this tissue, since the MDA levels in this group were significantly lower compared to the other treatments (HDS = 5.69 ± 1.23 nmol/L; CA = 7.42 ± 1.11 nmol/L and ZBS-CA/Ch = 2.22 ± 0.52 nmol/L).

#### 2.5.2. Antiedematogenic and Antioxidant of ZBS-CA/Ch in Carrageenan-Induced Paw Edema Model

As shown in Figure 6A, the development of edema induced by carrageenan started soon after the subcutaneous injection and the paw edema reached a maximum value in 90 min (4.894 ± 0.365 mm) and remained elevated until at least 240 min. Regarding treatments, HDS (4 mg/Kg), CA (10 mg/Kg) and ZBS-CA/Ch (50 mg/Kg) inhibited paw edema from 30 to 240 min (Phase 1 -AUC 30–120min and Phase 2-AUC 150–240 min) after carrageenan induction (CG: 1447 ± 112.3%; HDS: 172.8 ± 14.87%; CA: 177.9 ± 22.22% and ZBS-CA/Ch: 86.39 ± 7.46%) (Figure 6(B.1)).

It is important to note that the ZBS-CA/Ch nanoparticle was able to maintain edema inhibition for at least 240 min after stimulation (Figure 6(C.1–C.3)), showing a better anti-inflammatory action compared to the CA group, and similar action to the corticoid control (GC: 1502 ± 134.1%; HDS: 172.8 ± 3.08%; CA: 241.7 ± 17.92% and ZBS-CA/Ch: 132 ± 8.91%) (Figure 6(B.2)).

Regarding the oxidative parameters, 240 min after the induction of paw edema, carrageenan did not change the TEAC (Figure 6D), but reduced the levels of NO and MDA in the paw compared to the CG group. In this study, all treatments inhibited NO production (CG: 266.7 ± 5.65 μM/L; HDS: 3.08 ± 1.58 μM/L; CA: 9.35 ± 0.35 μM/L; ZBS-CA/Ch: 10.48 ± 1.025 μM) (Figure 6E) and MDA (CG: 22.71 ± 2.43 nMol/L; HDS: 5.69 ± 1.25 nM/L; CA: 7.42 ± 1.11 nM/L; ZBS-CA/Ch: 2.23 ± 1.02 nM/L) (Figure 6F), mainly the ZBS-CA/Ch group, which significantly inhibited lipid peroxidation (MDA) compared to the CA group, showing better antioxidant action (Figure 6F).

#### 2.5.3. ZBS-CA/Ch Not Produce Systemic Toxicity in Dextran and Carrageenan-Induced Paw Edema Models

The results of the oxidative balance in the liver 240 min after injection of dextran with treatment with CA did not change levels of TEAC and NO (Figure 7A,B), however, although ZBS-CA/Ch not alter TEAC levels in the liver (Figure 7A), it was able inhibited the production of NO (Figure 7B) and MDA in the liver induced by the stimuli better than CA (*p* = 0.0115) (Figure 7C).

In the carrageenan-induced paw edema model, CA treatment also did not change levels of TEAC, however nanoparticle of CA (ZBS-CA/Ch) reduced levels of TEAC compared CA group (*p* = 0.0105) (Figure 8A). On the other hand, both treatments were able to inhibit the production of NO induced by the stimuli (Figure 8B) and reduced lipid peroxidation (Figure 8C). In this study, ZBS-CA/Ch proved its better antioxidant action in relation to CA in carrageenan-induced paw edema than CA (*p* = 0.0062) (Figure 8C).

In the present study, in paw edema model, our data showed an excellent anti-edematogenic and antioxidant action of CA in free form and in the ZBS-CA/Ch nanoparticle, but the nanoparticles formulation proved to be more effective for a longer time in dextran-induced paw edema model. In this sense, several studies have already reported the activities of CA, a hydroxycinnamic acid, known to have cardioprotective action [2], antioxidant [38], antitumor [22,39], anti-inflammatory [40], antimicrobial [41] among others. However, the present study shows for the first time that ZBS-CA/Ch nanoparticles show better anti-inflammatory and antioxidant action in vivo.

In the dextran-induced paw edema model, in the initial phase (0–60 min), CA has the ability to prevent mast cell degranulation and consequently the release of mediators such as histamine [2]. Subsequently, in the second phase (60–240 min), CA inhibits the release of arachidonic acid from the cell membrane and consequently inhibits cyclooxygenase-2 (COX-2) and lipoxygenase (LOX) enzymes, especially 5-LOX [23,42]. According to Bare et al., (2019) [43], CA blocks phosphorylation of JNK, p38 and ERK, which inhibits COX-2 and consequently the conversion of PG2 to PH2, which reduces the production of superoxide and lipid peroxidation. The same was observed in our study, CA showed anti-inflammatory effect in the second phase of dextran-induced edema and in the parameters of oxidative stress, it was observed that the ZBS-CA/Ch nanoparticles prolonged the antioxidant activity of CA, reducing the lipid peroxidation in the paw.

In relation to the carrageenan, is a high molecular weight polysaccharide used in acute and chronic inflammatory processes [44]. This polysaccharide induces inflammatory process with systemic changes, and is subdivided into two phases associated with mediators, the first phase (30–120 min) is mediated by the release of histamine and serotonin, while the second (150–360 min) consists of an increase in prostaglandins E2, cytokines such as IL-1, IL-6 and TNF-α and reactive species such as NO [45,46,47]. According to Borthakur et al., 2012, the inflammatory process induced by carrageenan is mediated by the TLR-4/NF-κB signaling pathway, which is associated with increased expression of pro-inflammatory mediators, including cytokines and NO [48,49]. Studies shown that CA can modulate the inflammatory and antioxidant response through the modulation of the TLR4/TRIF/SYK/EROs signaling pathway [20,49]. In this regard, it was possible to observe the antiedematogenic, anti-inflammatory, and antioxidant effects of CA during the time evaluated.

In this study, the edema induced by dextran and carrageenan lead to an increase in systemically reactive species, and our data also shows the systemic inhibitory effect of CA and ZBS-CA/Ch nanoparticles on the production of NO induced by Dextran or Carrageenan. In addition, ZBS-CA/Ch nanoparticles were able to protect animals from oxidative damage in both models. An important biological action of CA is its potent antioxidant activity, capable of preventing the production of reactive oxygen species including species such as O_2_^−^, OH^−^, H_2_O_2_ preventing cell damage and tissue [50].

Another study in a carrageenan-induced paw edema model showed that caffeic acid phenethyl ester (CAPE), a CA derivative, at doses of 10 and 30 mg/Kg, can regulate the levels of heme oxygenase-1 (HO-1) enzymatic activity responsible for protecting against oxidative damage, with increased expression of the nuclear regulatory factor erythroid 2-related factor gene (Nrf2) [51]. The Nrf2 gene plays a key role in the regulation of expression of genes responsible for encoding antioxidant and anti-inflammatory proteins [52].

## 3. Materials and Methods

### 3.1. Chemicals and Reagents

All reagents were, purchased from Sigma-Aldrich-Brazil and used as received: Chitosan (Ch) of the low molecular weight, caffeic acid (CA, C_9_H_8_O_4_), zinc nitrate hexahydrate, Zn(NO_3_)_2_·6H_2_O (98%), metallic sodium, Na(s) (99.8%), ethyl alcohol, CH_3_CH_2_OH (99.5%), dimethyl sulfoxide-DMSO, (CH_3_)_2_SO (≥99.7%), phosphate buffer (98%). Deionized water was decarbonated before synthesis. Acetic acid solution for HPCL. Overall, all reagents used in this study were analytical grade.

### 3.2. Synthesis

The coprecipitation method was used in the synthesis process of efficient drug delivery systems. Zinc basic salt (ZBS, hereafter named) was prepared as previously reported [27]. The pristine ZBS was prepared by slowly adding 0.8 M NaOH (aq) to 3.75 M Zn(NO_3_)_2_·6H_2_O (aq) solution at room temperature with vigorous stirring and soon after, the white precipitate was immediately centrifugated and washed thoroughly with water and then dried at room temperature. For the intercalation of CA into the ZBS host 30 mL of water/methanol (5/1 *v*/*v*) was added to a three-neck flask containing CA, producing sodium caffeilate salt (C_9_H_7_O_4_^−^Na^+^). Soon after, a mixed aqueous solution (10.0 mL) containing Zn(NO_3_)_2_·6H_2_O was slowly added to the C_9_H_7_O_4_^−^Na^+^ solution. To ensure the dissolution of CA, the pH was adjusted to approximately 7.0 using 1.0 M NaOH solution. The addition was performed in the dark and under vigorous stirring in a nitrogen atmosphere to protect the reagents from light and oxygen. During the procedure, the pH was maintained by the continuous addition of a 1.0 M NaOH solution. The resulting suspension was immediately centrifuged and washed thoroughly with water and dried in a vacuum desiccator. 0.8 g of Chitosan (Ch) [53] was used for ZBS-Ch synthesis. The Ch was dissolved in aqueous solution of 100 mL (1%) acetic acid, where the pH of the solution was adjusted at 7.0 by adding 1.0 M NaOH. Then, this solution was kept in vigorously stirred under a nitrogen atmosphere at room temperature for 1 h. This solution was added dropwise to a suspension containing 0.4 g of ZBS in 50 mL of deionized water. Its pH was adjusted to 7.0 by the controlled addition of 1.0 M NaOH under magnetic stirring and the nitrogen flow at room temperature during a time interval. The solid was isolated by centrifugation and washed thoroughly three times with aqueous ethanol solution (50 vol. %) and then dried in a vacuum desiccator. The new efficient drug delivery system (ZBS-CA/Ch) is presented. In the present method, Ch was vigorously stirred in a solution of acetic acid under a nitrogen atmosphere at stirred at room temperature at a predetermined time interval. The pH of the solution was kept nearly at 7.0 by adding 1.0 M NaOH. This solution was added dropwise to a suspension containing ZBS-CA in distilled water. Its pH was adjusted to 7.0 by the controlled addition of 1.0 M NaOH under magnetic stirring and a nitrogen flow at room temperature. The solid was isolated by centrifugation and washed with aqueous ethanol solution and then dried in a vacuum desiccator.

### 3.3. Sample Characterization

X-ray powder diffraction (XRD) of all the samples was collected from 2 to 80° (2θ) with a step of 0.04° and an effective acquisition time of 2.8 s per step using a Bruker D8 Advance diffractometer with Bragg-Brentano geometry, LynxEye PSD detector, and CuK C_u_K_α_ radiation (λ = 1.5418 Å). The XRD data were extracted by the PEAKOC, Version 1.0, https://www.esrf.fr (accessed on 20 September 2022) [54] using a split pseudo-Voigt function [55] to fit the experimental profiles. FT-IR spectra of the powder samples were scanned (400–4000 cm^−1^) on a Bruker (Vertex 70v) spectrophotometer with a resolution of 2 cm^−1^. Each spectrum was the average of 100 successive scans.

The solid state UV–Vis spectra for pure CA, pure Ch and ZBS-CA/Ch powder were obtained in a range from 200 nm to 900 nm with BaSO_4_ background by using a UV-Vis spectroscope (Shimadzu UV-2600 model). In addition, CA quantification was performed after the destruction of the ZBS-CA structure in an acidic medium. Measurements were carried out using a UV-Vis spectroscope (Shimadzu UV-2600 model) as described in our previous work [56]. One milligram of ZBS-CA sample was dissolved in 0.5 mL of ethanol and 0.5 mL of 0.1 M HCl solution, with a successive dilution using a phosphate buffer at pH 7.40 ± 0.02 in a 10 mL volumetric flask. The concentration of CA in the resulting solution was determined by measuring the absorbance at λ_max_ = 325 nm. The concentration of CA was calculated by regression analysis according to a calibration curve obtained from a series of standard solutions of pure CA.

### 3.4. Caffeic Acid Release Studies

The release of caffeic acid was performed as described in our previous work [57]: the ZBS-CA or ZBS-CA/Ch samples (10 mg) were immersed in phosphate-buffered saline at pH 4.8 and 7.4 (100 mL) with mild shaking at 150 rpm; the temperature was maintained at 37 ± 0.5 °C. An aliquot of 5 mL was taken from the suspension from time to time to measure the release of caffeic acid into the solution. Aliquots were immediately replaced by an equal volume of SBF at 37 ± 0.5 °C to keep the temperature and volume constant. The content of caffeic acid in each aliquot was filtered and measured at an absorbance of 325 nm using a UV/Vis spectrophotometer (Shimadzu UV-2600 model) at λ_max_ = 325 nm according to a previously determined calibration curve (y = 0.0426x + 0.0229; *r*^2^ = 0.9999). The percentage released at each time was expressed as a fraction of the total amount of CA. Drug release was monitored for 10 h, and the CA concentration was collected as the average of 3 measurements. All recordings were within the range of the calibration curve. Moreover, in order to obtain more information on the release behavior of CA from the ZBS-CA and ZBS-CA/Ch samples, five kinetic models were applied: zero order, first order, pseudo-second order, Korsmeyer-Peppas and Higuchi model (Equations (1)–(5)) [31,58,59].
𝑞 = 𝑘𝑡 + 𝑐(1)
ln(𝑞𝑒−𝑞𝑡) = ln𝑞𝑡 − 𝑘𝑡(2)
(3)$$\frac{1}{q_{t}}=\frac{1}{kq_{e}^{2}}+\frac{1}{q_{e}}$$
(4)$$\frac{t}{q_{\infty }}=kt^{n}$$
(5)$$q_{t}=k\sqrt{t}$$
where $q_{e}$ is the equilibrium release amounts, $q_{t}$ the release amounts at any time (t), k is the rate constant, and *c* an arbitrary constant.

### 3.5. Animals Model

#### 3.5.1. Ethics Statement and Animals

This study was carried out in strict accordance with the recommendations of the Guide for the Care and Use of Laboratory Animals of the Brazilian National Council of Animal Experimentation http://www.sbcal.org.br/ (accessed on 30 August 2018) and the NIH Guidelines for the Care and Use of Laboratory Animals. The institutional Committee for Animal Ethics of the Federal University of Pará (CEUA, Protocol: 9731050718) approved all the procedures used in this study. A total of 40 healthy males and females (*Rattus novergicus*) of *Wistar* lineage, were obtained from the Central Animal Facility of Federal University of Pará and kept in cages under controlled conditions of temperature (22 ± 3 °C), light (12 h light/dark cycle) with food and water *ad libitum*, and acclimatized conditions for 3 days before use. On the day of the experiment, animals were randomly allocated in groups of four inside cages.

#### 3.5.2. Paw Edema Model

Paw edema was induced by subcutaneous (s.c.) injection of dextran (1% *w*/*v*, 100 μL) or carrageenan (1% *w*/*v*, 100 μL) into the animals right hind paws according to Winter et. al. (1962) [60] and Ghorbanzadeh et al. (2015) [61]. Inflammation-induced edema was calculated by measuring the changes in paw thickness with a digital caliper (Absolute-Série 500, Mitutoyo, Japan) just prior to (t_0min_) and every 30 min until 4 h after (t_240min_) the dextran or carrageenan injection. The areas under the time course curves (AUC) were calculated using the trapezoidal rule [62,63].

#### 3.5.3. Design of In Vivo Experiments

The animals were randomized into two groups (dextran and carrageenan) with 20 animals each. The dextran and carrageenan groups were separated into four groups each according to the treatment.

*Carrageenan (CG)* or *Dextran (DEX):* 30 min after dextran (1% *w*/*v*, 100 μL) or carrageenan injection (1% *w*/*v*, 100 μL, s.c.) physiological saline (100 μL, s.c.) was administered into the right hind paw (n = 5 per group).

*Hydrocortisone Group (HDS):* 30 min after dextran (1% *w*/*v*, 100 μL) or carrageenan injection (1% *w*/*v*, 100 μL, s.c.) physiological saline (100 μL, s.c.) and 4 mg/Kg body weight of Hydrocortisone were administered into the right hind paw (n = 5 per group).

*Caffeic acid (CA):* 30 min after dextran (1% *w*/*v*, 100 μL) or carrageenan injection (1% *w*/*v*, 100 μL, s.c.) physiological saline (100 μL, s.c.) and 10 mg/Kg body weight of Caffeic acid (CA) were administered into the right hind paw (n = 5 per group).

*Nanoparticles based on Chitosan/Zinc basic salt with caffeic acid drug Group (ZBS-CA/Ch):* 30 min after dextran (1% *w*/*v*, 100 μL) or carrageenan injection (1% *w*/*v*, 100 μL, s.c.) and 50 mg/Kg body weight of ZBS-CA/Ch were administered into the right hind paw (n = 5 per group).

The nanoparticle dose used in the treatment of paw edema was determined from the percentage of (CA) present in the compound, so that 10 mg/kg of the compound would be (CA) and the remaining 40 mg/kg of ZBS-CH complex. The animals were evaluated for 240 min and posteriorly were euthanized to collect paw and liver to the determination of the oxidative stress state.

#### 3.5.4. Total Evaluation of Trolox Equivalent Antioxidant Capacity (TEAC)

The total antioxidant capacity (TAC) of paw and liver specimens 240 min post-dextran- or -carrageenan-induced paw edema was evaluated via Trolox ((±)-6-Hydroxy-2,5,7,8-tetramethylchromane-2-carboxylic acid; (Sigma-Aldrich, Co,3050 Spruce St., St Louis, MO, USA) equivalent antioxidant capacity assay (TEAC), which provides relevant information that may effectively describe the dynamic equilibrium between pro-oxidant and antioxidant compounds. In this assay, 2,2′-Azino-bis (3-ethylbenzothiazoline-6-sulfonic acid) diammonium salt (ABTS) (Sigma Aldrich) was incubated with potassium persulfate (Sigma Aldrich) to produces ABTS∙+, a green/blue chromophore. The inhibition of ABTS∙+ formation by antioxidants in the samples was expressed as Trolox equivalents, determined at 740 mm using a calibration curve plotted with different amounts of Trolox (Sigma Aldrich) [64].

#### 3.5.5. Determination of Nitric Oxide (NO) Production

The nitrite (NO_2_) was estimated calorimetrically in paw and liver 240 min post dextran or carrageenan-induced paw edema based on reduction of nitrate to nitrite using the Griess method. Nitrite level was determined in 100 μL of samples (serum and lavage peritoneal) incubated with an equal volume of Griess reagent for 10 min at room temperature. The absorbance was measured at 550 nm and calculated from a standard curve with sodium nitrite expressed per μM/mL [47].

#### 3.5.6. Lipid Peroxidation

Lipid peroxidation was measured in paw and liver specimens 240 min post-dextran- or -carrageenan-induced paw edema as an indicator of oxidative stress, using the thiobarbituric acid-reactive substances (TBARS) assay [65,66]. Briefly, samples were mixed with 0.05 M trichloroacetic acid (TCA) and 0.67% thiobarbituric acid (TBA; Sigma-Aldrich, St. Louis, MO, USA) in 2 M sodium sulfate and heated in a water bath at 94 °C for 90 min. The chromogen formed was extracted in n-butanol and measured at 535 nm. An MDA standard solution was used to construct a standard curve against which unknown samples were plotted. Results are expressed as malondialdehyde equivalents in nmol/L.

### 3.6. Statistical Analysis

Results are expressed as mean ± SD from at least 5 animals per group and statistical analysis was performed using one-way analysis of variance (ANOVA) followed by Tukey’s test for the comparison of pairs of means and Pearson’s correlation tests, considered statistically significant for *p* ≤ 0.05.

## 4. Conclusions

In this study, the intercalation of the caffeilate anion into interlayer space of zinc basic salts (ZBS) was carried out by coprecipitation method. Then, ZBS-CA nanohybrid material were used to prepare the ZBS-CA/Ch by coating of ZBS-CA complex with Chitosan, and characterized by X-ray diffraction (XRD), Fourier transform infrared (FTIR), and UV-visible (UV-vis) spectroscopy. The data analysis reveals the percentage of 46% and 28% of the CA into ZBS-CA and ZBS-CA/Ch, respectively. After 10 h, the release of caffeate anions from ZBS-CA is 77% and 41% at pH 4.8 and 7.4, respectively. For ZBS-CA/Ch system, it was observed a different trend (after the same amount of time), where CA release correspond to 46% and 23% at pH 4.8 and 7.4, respectively. Furthermore, pseudo-second order is the best kinetics model to explain the release of CA anion from the interlayer spaces ZBS, with a linear correlation coefficient of R^2^ ≥ 0.99 at pH 4.8 and pH 7.4. Overall, nanoparticle containing CA (ZBS-CA/Ch) shows better anti-inflammatory and antioxidant actions in the two inflammatory phases of carrageenan, for a longer time compared to the CA group, as well as leading to a balance in the oxidative process induced by carrageenan in the paw. ZBS-CA/Ch nanoparticle slowly releases caffeate anions in the tissue, leading to a prolongation of CA-induced anti-edematogenic and anti-inflammatory activities, as well as improving its inhibition or sequestration antioxidant action of reactive species.

## Figures and Tables

**Figure 1 molecules-28-04973-f001:**
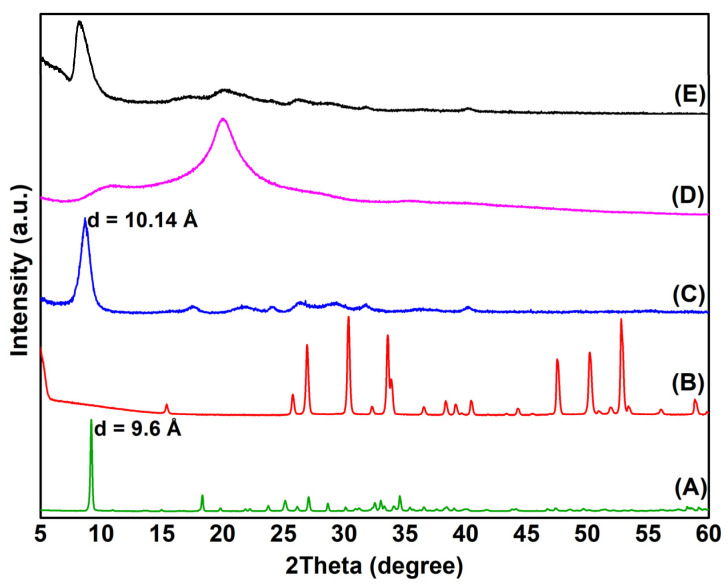
X-ray powder diffraction patterns of the ZBS-NO_3_ (**A**), pure CA (**B**), ZBS-CA (**C**), pure Ch (**D**) and ZBS-CA/Ch (**E**) intercalation compound.

**Figure 2 molecules-28-04973-f002:**
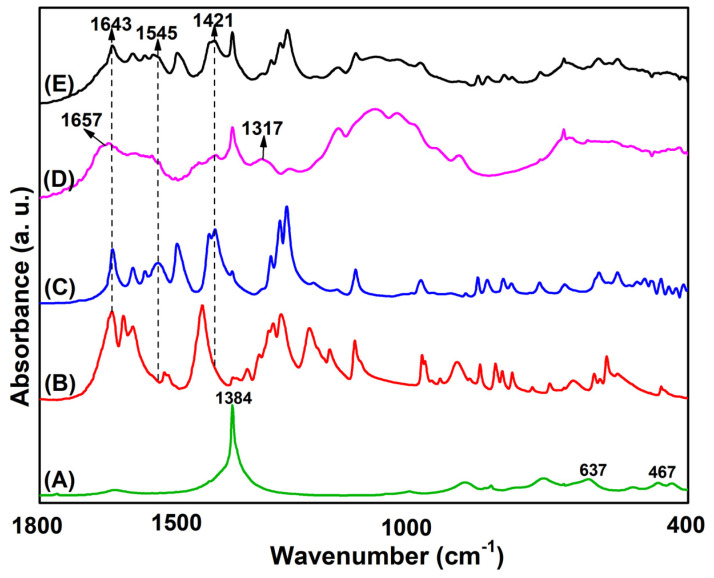
FTIR spectra of (**A**) pristine ZBS (Zn-NO_3_), (**B**) pure CA, (**C**) ZBS-CA, (**D**) pure Ch and (**E**) ZBS-CA/Ch samples.

**Figure 3 molecules-28-04973-f003:**
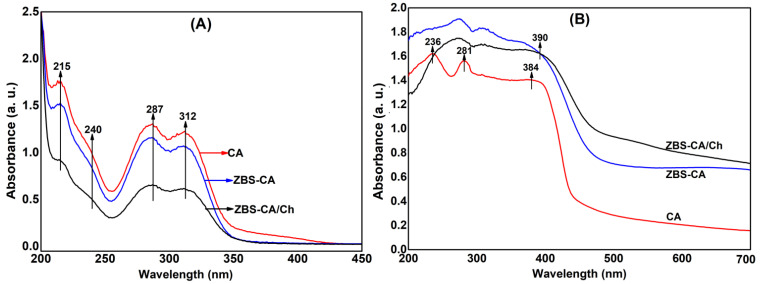
(**A**) Solution UV-vis spectra of the pure CA (red line), released CA from the ZBS-CA (blue line) and released CA from the ZBS-CA/Ch (black line). (**B**) Solid-state UV-vis spectra of the pure CA (red line), released CA from the ZBS-CA (blue line) and released CA from the ZBS-CA/Ch (black line).

**Figure 4 molecules-28-04973-f004:**
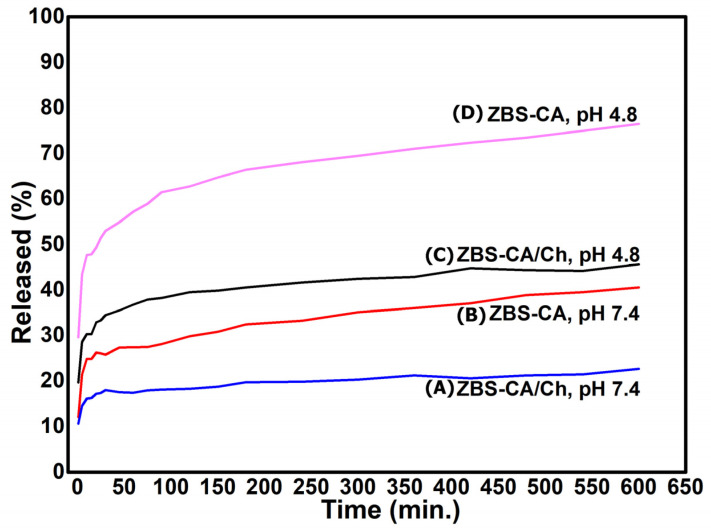
Release profiles for caffeic acid from the *nanoparticles* based on chitosan/zinc basic salt with caffeic acid drug at pH 7.4 and pH 4.8: (**A**) ZBS-CA/Ch (pH 7.4); (**B**) ZBS-CA (pH 7.4); (**C**) ZBS-CA/Ch (pH 4.8) and (**D**) ZBS-CA (pH 4.8).

**Figure 5 molecules-28-04973-f005:**
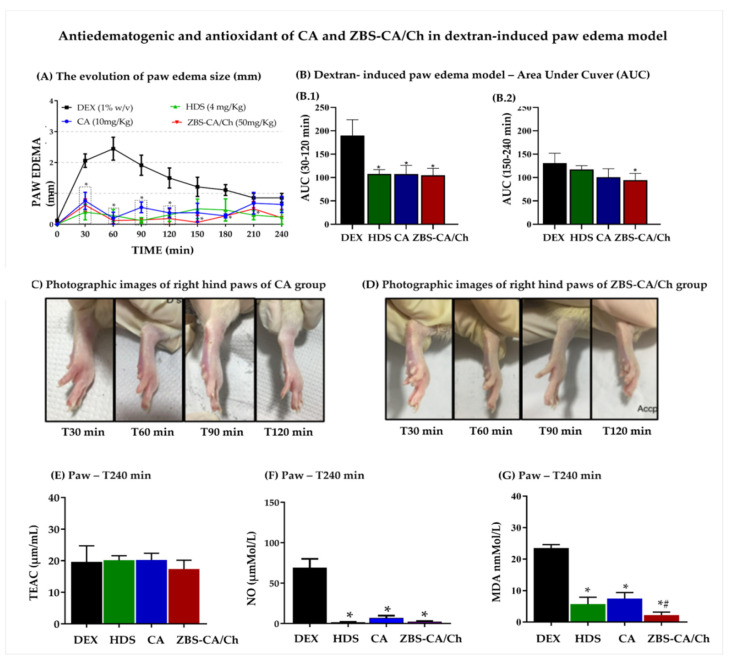
Effect antiedematogenic and antioxidant of CA (10 mg/Kg) and ZBS-CA/Ch (50 mg/Kg) in dextran-induced paw edema model. (**A**) The evolution of paw edema size (mm) in different times (T0–240 min) after dextran injection. (**B**) Dextran induced paw edema model—Area Under Cuver (AUC). (**B.1**) AUC graph in minute 30–120 after dextran induced paw edema in association between measurement time (minute) to rat paw thickness (mm). (**B.2**) AUC graph in minute 120–240 after dextran induced paw edema in association between measurement time (minute) to rat paw thickness (mm). (**C**) Photographic images of right hind paws of CA group in different times (T30, 60, 90 and 120) after dextran injection (**D**) Photographic images of right hind paws of ZBS-CA/Ch group in different times (T30, 60, 90 and 120 min) after dextran injection (**E**) TEAC of rats with dextran-induced paw inflammation 240 min after induction. (**F**) NO levels of rats with dextran-induced paw inflammation 240 min after induction. (**G**) MDA (lipid peroxidation) of rats with dextran-induced paw inflammation 240 min after induction. Data were expressed as mean ± SD (n = 5 per group). Statistically significant differences between HDS, CA and ZBS-CA/Ch group vs. DEX group at * *p* < 0.05. Statistically significant differences between CA vs. ZBS-CA/Ch group at ^#^
*p* < 0.05. Legend: CA: caffeic acid, DEX: Dextran, HDS: hydrocortisone, MDA: malondialdehyde, NO: nitric oxide, TEAC: Total Evaluation of Trolox Equivalent Antioxidant Capacity, ZBS-CA/Ch: Nanoparticles based on Chitosan/Zinc basic salt with caffeic acid drug group.

**Figure 6 molecules-28-04973-f006:**
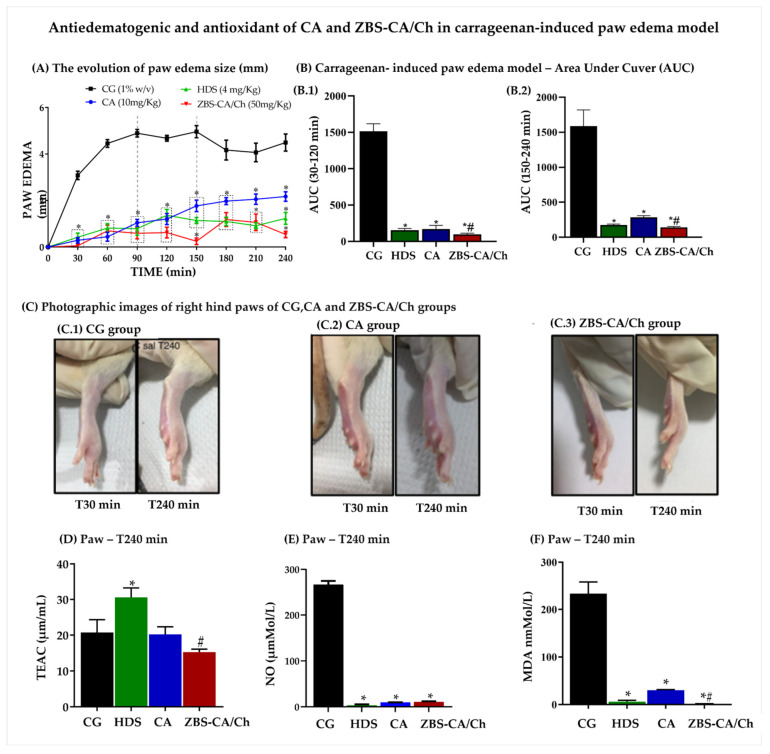
Effect antiedematogenic and antioxidant of CA (10 mg/Kg) and ZBS-CA/Ch (50 mg/Kg) in carrageenan-induced paw edema model. (**A**) The evolution of paw edema size (mm) in different times (T0–240 min) after carrageenan induction (**B**) Carrageenan- induced paw edema model—Area Under Curve (AUC). (**B.1**) AUC graph in minute 30–120 after carrageenan induced paw edema in association between measurement time (minute) to rat paw thickness (mm). (**B.2**) AUC graph in minute 120–240 after carrageenan induced paw edema in association between measurement time (minute) to rat paw thickness (mm). (**C**) Photographic images of right hind paws of (**C.1**) CG, (**C.2**) CA and (**C.3**) ZBS-CA/Ch group T30 and T240 min after carrageenan injection. (**D**) TEAC of rats with carrageenan-induced paw inflammation 240 min after induction. (**E**) NO levels of rats with carrageenan-induced paw inflammation 240 min after induction. (**F**) MDA of rats with carrageenan-induced paw inflammation 240 min after induction. Data were expressed as mean ± SD (n = 5 per group). Statistically significant differences between HDS, CA and ZBS-CA/Ch group vs. CG group at * *p* < 0.05. Statistically significant differences between CA vs. ZBS-CA/Ch group at ^#^
*p* < 0.05. Legend: CA: caffeic acid, CG: carrageenan, HDS: hydrocortisone, MDA: malondialdehyde, NO: nitric oxide, TEAC: Total Evaluation of Trolox Equivalent Antioxidant Capacity, ZBS-CA/Ch: Nanoparticles based on Chitosan/Zinc basic salt with caffeic acid drug group.

**Figure 7 molecules-28-04973-f007:**
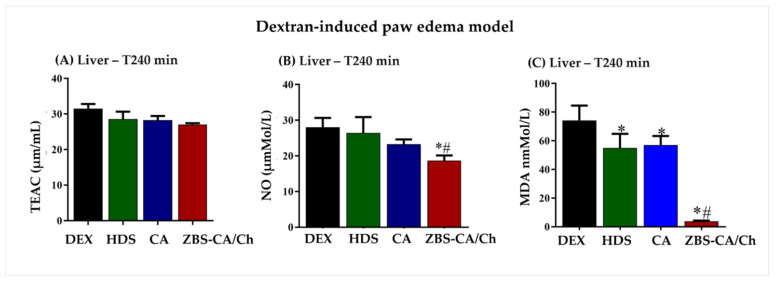
ZBS-CA/Ch not produce systemic toxicity 240 min after dextran induced paw edema (**A**) TEAC in the liver 240 min after injection of dextran (**B**) MDA in the liver 240 min after injection of dextran (**C**) NO in the liver 240 min after injection of dextran. Data were expressed as mean ± SD (n = 5 per group). Statistically significant differences between HDS, CA and ZBS-CA/Ch group vs. DEX group at * *p* < 0.05. Statistically significant differences between CA vs. ZBS-CA/Ch group at ^#^
*p* < 0.05. Legend: CA: caffeic acid, DEX: dextran, HDS: hydrocortisone, MDA: malondialdehyde, NO: nitric oxide, TEAC: Total Evaluation of Trolox Equivalent Antioxidant Capacity, ZBS-CA/Ch: Nanoparticles based on Chitosan/Zinc basic salt with caffeic acid drug group.

**Figure 8 molecules-28-04973-f008:**
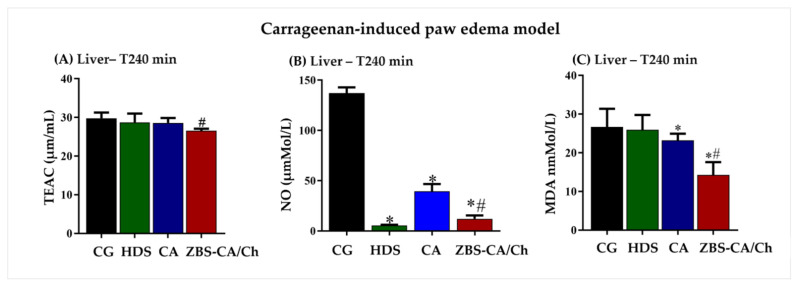
ZBS-CA/Ch not produce systemic toxicity 240 min after carrageenan-induced paw edema. (**A**) TEAC in the liver 240 min after injection of carrageenan (**B**) MDA in the liver 240 min after injection of carrageenan (**C**) NO in the liver 240 min after injection of carrageenan. Data were expressed as mean ± SD (n = 5 per group). Statistically significant differences between HDS, CA and ZBS-CA/Ch group vs. CG group at * *p* < 0.05. Statistically significant differences between CA vs. ZBS-CA/Ch group at ^#^
*p* < 0.05. Legend: CA: caffeic acid, CG: carrageenan, HDS: hydrocortisone, MDA: malondialdehyde, NO: nitric oxide, TEAC: Total Evaluation of Trolox Equivalent Antioxidant Capacity, ZBS-CA/Ch: Nanoparticles based on Chitosan/Zinc basic salt with caffeic acid drug group.

**Table 1 molecules-28-04973-t001:** Correlation coefficient (R^2^) obtained by fitting caffeic acid release data from ZBS-CA and ZBS-CA/Ch samples into phosphate-buffered saline at pH 7.4 and pH 4.8.

Sample	pH	Saturation Release (%)	Kinetic Models						
			Korsmeyer-Peppas	Higuchi	Zero-Order	First-Order	Simulation of the Kinetic Pseudo-Second Order Model		
							R^2^	Stage I	Stage II
**ZBS-CA**	4.8	77.0	0.978	0.953	0.878	0.918	0.997	0.996	0.998
**ZBS-CA/Ch**		46.0	0.959	0.919	0.934	0.959	0.998	0.997	0.999
**ZBS-CA**	7.4	41.0	0.915	0.985	0.733	0.786	0.992	0.998	0.995
**ZBS-CA/Ch**		23.0	0.915	0.936	0.788	0.819	0.996	0.999	0.996

## Data Availability

Not applicable.

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
