# Peer review of "Caffeic Acid-Zinc Basic Salt/Chitosan Nanohybrid Possesses Controlled Release Properties and Exhibits In Vivo Anti-Inflammatory Activities"

_molecules, 2023, doi:10.3390/molecules28134973_

Round 1

Reviewer 1 Report

1.     I cannnot understand the Fig.4, why the author only conduct the releasing profile at pH 4.8? not 5.6 or PBS?

2.     Also, the Fig. 4 is not well, how about is the error analysis? It should be worked.

3.     Please provide the TEM and SEM for all the samples.

4.     The introduction is badly, it should be written.

5.     “Thus, the search for new formulations where the drug is carried is safe and effective for new therapies for various treatments.” This should be added the refs, such as J. Control. Release, 354(2023)615–625; Molecules, 27(2022) 7166; Dalton Transactions, 2022, 51, 14817-14832 and J. Mater. Chem. B., 2022, 10, 5105 - 5128

revision

Author Response

Dear,

Attached is the answer to the questions.

Thank you for your consideration of this manuscript.

Sincerely,

Reviewer 2 Report

The manuscript by Ferreira Meneses et al. entitled "A Drug Delivery System Based on Chitosan/Zinc Basic Salt and Caffeic Acid with Anti-edemato-genic Activity" (molecules-2410244) describes a drug delivery system made of Chitosan (Ch) and Zinc Basic Salt (ZBS) for delivery of caffeic acid (CA) to cure the edema-associated symptoms. The micro composites (ZBS-Ca/Ch) were characterized by X-ray diffraction, Fourier transform infrared (FT-IR), and UV-visible spectroscopy. Moreover, a drug release study was performed. The micro composites were also tested in vivo in the rat edema model induced by dextran or carrageenan.

It is a potentially interesting study but requires greater transparency in the presentation of the data. The manuscript was not carefully prepared, and the descriptions of the pictures contain several errors. English correction is highly recommended (especially in the Introduction section). The authors should try to avoid long sentences.

To improve the manuscript, please accept some comments.

Major comments:

1.      The title needs to be more adequate for the presented study. The manuscript presents a study on a drug delivery system based on Chitosan/Zinc Basic Salt that delivers Caffeic Acid. Please modify accordingly.

2.      The obtained material was named micro composites and is quite extensively characterized in composition. However, the graphical abstract showing micro composites' structure needs to be clarified. Could you show the SEM images of the nanoparticles? How does micro composites (or nanoparticles?) morphology look like? What is the size of the particles? Or Size distribution? Is it injectable material? Please include additional data.

3.      The authors used 10 mg/kg body weight of Caffeic acid (CA) and 50 mg/kg of ZBS-CA/Ch) in the animal study. What was the loading efficiency of CA to particles? Are the applied doses equivalent? Could you explain/discuss the applied doses of the drug?

4.      Figure 3; Please carefully analyze the figure and the figure's description. The samples are mistaken. Please correct. 

5.      Figure 4; According to the text and figure's description, one expects to see two figures 4A and B). It looks like the second part of the figure is missing. Why the release of CA at pH 7.8 was not presented? Various fonts are used in the figure's description. Please correct.

6.      The mode of action of dextran or carrageenan is mentioned at least three times in the text. Please correct it.

7.      Figure 5. Please include in the figure description all the necessary information to understand the figure. In the present form, the figure is not understandable. The quality of the inscriptions in the pictures could be at better resolution. 

8.      In the text (during the description of the results), please include the number of pictures to which the text refers. In the present form, it is hard to follow the manuscript.

9.      Please standardize the way of presentation of references in the text (page 2) or units (kg vs.Kg)

10.  Figure 6; the chapter describing Figure 6 was mistakenly named "4. Discussion." Please correct it accordingly.

11.  Figure 6. Please include all the necessary information to understand the figure in the figure description. Do not copy from Figure 5. As the reviewer understands, Figure 6 shows the results in the carrageenan-induced edema model.

12.  The page numbers need to be corrected. 

13.  Figure 7. Please include in the figure description all the necessary information to understand the figure. Divide the picture into dextran- and carrageenan-induced edema models.

 English correction is highly recommended (especially in the Introduction section). For the clarity of the presentation, authors should avoid long sentences.

Author Response

(The authors gave the same response as above.)

Round 2

Reviewer 1 Report

accept

Reviewer 2 Report

The authors responded to most of the comments accordingly.

Thank you for changing the title; however, according to the results, the most promising composition was “Caffeic acid- Chitosan-zinc basic salt nanohybrid possesses controlled release properties and exhibits in vivo anti-inflammatory activities.”

Moreover, please double-check and standardize the way of presentation of units (kg vs.Kg).

The reference list needs significant revision. Compared with the previous reference list, it has been changed completely. It looks like it is from the other article. We can find plenty of articles concerning batteries...

Examples:

21. Liu, Y.; Lei, T.; Li, Y.; Chen, W.; Hu, Y.; Huang, J.; Chu, J.; Yan, C.; Wu, C.; Yang, C. Entrapment of polysulfides by a BiFeO3/TiO2 heterogeneous structure on separator for high-performance Li–S batteries. J. Power Sources 2023, 556, 232-501.

22. Wang, W.; Xi, K.; Li, B.; Li, H.; Liu, S.; Wang, J.; Zhao, H.; Li, H.; Abdelkader, A.M.; Gao, X.; et al. A Sustainable Multipurpose Separator Directed Against the Shuttle Effect of Polysulfides for High‐Performance Lithium–Sulfur Batteries. Adv. Energy Mater. 2022, 12, 220-160.

23. Gao, Z.; Xue, Z.; Miao, Y.; Chen, B.; Xu, J.; Shi, H.; Tang, T.; Zhao, X. TiO2@Porous carbon nanotubes modified separator as polysulfide barrier for lithium-sulfur batteries. J. Alloys Compd. 2022, 906, 164-249.

24. Han, Z.; Ren, H.-R.; Huang, Z.; Zhang, Y.; Gu, S.; Zhang, C.; Liu, W.; Yang, J.; Zhou, G.; Yang, Q.-H.; et al. A Permselective Coating Protects Lithium Anode toward a Practical Lithium–Sulfur Battery. ACS Nano 2023, 17, 4453–4462.

25. Castillo, J.; Coca-Clemente, J.A.; Rikarte, J.; Sáenz de Buruaga, A.; Santiago, A.; Li, C. Recent progress on lithium anode protection for lithium–sulfur batteries: Review and perspective. APL Mater. 2023, 11, 010-901.

26. Yan, Y.; Qin, H.; Wei, Y.; Yang, R.; Xu, Y.; Chen, L.; Li, Q.; Shi, M. A Mild Strategy to Strengthen Three Dimensional Graphene Aerogel for Supporting Sulfur as a Free-standing Cathode in Lithium-Sulfur Batteries. Bull. Korean Chem. Soc. 2018, 39, 605– 610.

27. Liu, Y.; Yao, M.; Zhang, L.; Niu, Z. Large-scale fabrication of reduced graphene oxide-sulfur composite films for flexible lithium-sulfur batteries. J. Energy Chem. 2019, 38, 199–206.

28. Yuan, N.; Deng, Y.-R.; Wang, S.-H.; Gao, L.; Yang, J.-L.; Zou, N.-C.; Liu, B.-X.; Zhang, J.-Q.; Liu, R.-P.; Zhang, L. Towards superior lithium–sulfur batteries with metal–organic frameworks and their derivatives. Tungsten 2022, 4, 269–283.

29. Huang, Y.-C.; Hsiang, H.-I.; Chung, S.-H. Investigation and Design of High-Loading Sulfur Cathodes with a HighPerformance Polysulfide Adsorbent for Electrochemically Stable Lithium–Sulfur Batteries. ACS Sustain. Chem. Eng. 2022, 10, 9254–9264.

30. Chen, Y.; Choi, S.; Su, D.; Gao, X.; Wang, G. Self-standing sulfur cathodes enabled by 3D hierarchically porous titanium monoxide-graphene composite film for high-performance lithium-sulfur batteries. Nano Energy 2018, 47, 331–339.

31. Liu, Y.; Liu, S.; Li, G.; Yan, T.; Gao, X. High Volumetric Energy Density Sulfur Cathode with Heavy and Catalytic Metal Oxide Host for Lithium–Sulfur Battery. Adv. Sci. 2020, 7, 1903693.

32. Xu, F.; Tang, Z.; Huang, S.; Chen, L.; Liang, Y.; Mai, W.; Zhong, H.; Fu, R.; Wu, D. Facile synthesis of ultrahigh-surface-area hollow carbon nanospheres for enhanced adsorption and energy storage. Nat. Commun. 2015, 6, 7221.

33. Chu, H.; Noh, H.; Kim, Y.-J.; Yuk, S.; Lee, J.-H.; Lee, J.; Kwack, H.; Kim, Y.; Yang, D.-K.; Kim, H.-T. Achieving threedimensional lithium sulfide growth in lithium-sulfur batteries using high-donor-number anions. Nat. Commun. 2019, 10, 188.

34. Tang, X.; Guo, X.; Wu, W.; Wang, G. 2D Metal Carbides and Nitrides (MXenes) as High‐Performance Electrode Materials for Lithium‐Based Batteries. Adv. Energy Mater. 2018, 8, 180-1897.

Please modify the reference list accordingly.

The reference list needs significant revision. Compared with the previous reference list, it has been changed completely. It looks like it is from the other article. We can find plenty of articles concerning batteries...